# `MiME`: Multilevel Medical Embedding of Electronic Health Records for Predictive Healthcare

**Edward Choi**[*]
Google Brain
edwardchoi@google.com

**Cao Xiao**
IBM Research
cxiao@us.ibm.com

**Walter F. Stewart**[†]
HINT Consultants
wfs502000@yahoo.com

**Jimeng Sun**
Georgia Institute of Technology
jsun@cc.gatech.edu

## Abstract

Deep learning models exhibit state-of-the-art performance for many predictive healthcare tasks using electronic health records (EHR) data, but these models typically require training data volume that exceeds the capacity of most healthcare systems. External resources such as medical ontologies are used to bridge the data volume constraint, but this approach is often not directly applicable or useful because of inconsistencies with terminology. To solve the data insufficiency challenge, we leverage the inherent multilevel structure of EHR data and, in particular, the encoded relationships among medical codes. We propose Multilevel Medical Embedding (`MiME`) which learns the multilevel embedding of EHR data while jointly performing auxiliary prediction tasks that rely on this inherent EHR structure without the need for external labels. We conducted two prediction tasks, *heart failure prediction* and *sequential disease prediction*, where `MiME` outperformed baseline methods in diverse evaluation settings. In particular, `MiME` consistently outperformed all baselines when predicting heart failure on datasets of different volumes, especially demonstrating the greatest performance improvement (15% relative gain in PR-AUC over the best baseline) on the smallest dataset, demonstrating its ability to effectively model the multilevel structure of EHR data.

## 1 Introduction

The rapid growth of electronic health record (EHR) data has motivated use of deep learning models and demonstrated state-of-the-art performance in diagnostics [26, 13, 12, 27], disease detection [14, 10, 17], risk prediction [20, 32], and patient subtyping [3, 6]. However, training optimal deep learning models typically requires a large volume (*i.e.* number of patient records and features per record) Most health systems do not have the data volume required to optimize performance of these models, especially for less common services (*e.g.* intensive care units (ICU)) or rare conditions.

External resources, particularly medical ontologies have been used to address data volume insufficiencies [12, 31, 7]. For example [12], latent embedding of a clinical code (e.g. diagnosis code) can be learned as a convex combination of the embeddings of the code itself and its ancestors on the ontology graph. However, medical ontologies are often not available or not directly applicable due to the nonstandard, or idiosyncratic use of terminology and complex terminology mapping from one health system's EHR to another. For example, many clinics still use their own in-house terminologies

---

[*]Work done at Georgia Institute of Technology.
[†]Work done at Sutter Health.

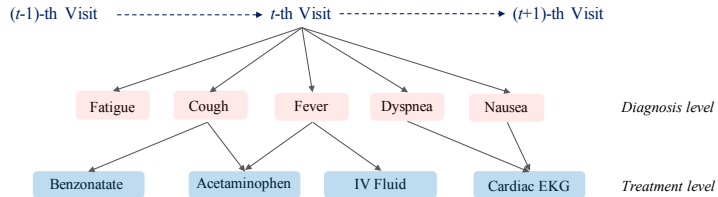

Figure 1: Symbolic representation of a single visit of a patient. Red denotes diagnosis codes, and blue denotes medication/procedure codes. A visit encompasses a set of codes, as well as a hierarchical structure and heterogeneous relations among these codes. For example, while both *Acetaminophen* and *IV fluid* form an explicit relationship with *Fever*, they also are correlated with each other as descendants of *Fever*.

for medications and lab tests, which do not conform with the standard medical ontologies such as Anatomical Therapeutic Chemical (ATC) Classification system and Logical Observation Identifiers Names and Codes (LOINC).

As an alternative, we explored how the inherent multilevel structure of EHR data could be leveraged to improve learning efficiency. The hierarchical structure of EHR data begins with the patient, followed by visits, then diagnosis codes within visits, which are then linked to treatment orders (*e.g.* medications, procedures). This hierarchical structure reveals influential multilevel relationships, especially between diagnosis codes and treatment codes. For example, a diagnosis *fever* can lead to associated treatments such as *acetaminophen* (medication) and *IV fluid* (procedure). We examine whether this multilevel structure could be leveraged to obtain a robust model under small data volume. To the best of our knowledge, none of the existing works leverage this multilevel structure in EHR. Rather, they flatten EHR data as a set of independent codes [18, 38, 11, 12, 14, 10, 13, 27, 2], which ignores hierarchical relationships among medical codes within visits.

We propose **M**ultilevel **M**edical **E**mbedding) (MiME) to simultaneously transform the inherent multi-level structure of EHR data into multilevel embeddings, while jointly performing auxiliary prediction tasks that reflect this inherent structure without the need for external labels. Modeling the inherent structure among medical codes enables us to accurately capture the distinguishing patterns of different patient states. The auxiliary tasks inject the hierarchical knowledge of EHR data into the embedding process such that the main task can borrow prediction power from related auxiliary tasks. We conducted two prediction tasks, *heart failure prediction* and *sequential disease prediction*, where MiME outperformed baseline methods in diverse evaluation settings. In particular, for heart failure prediction on datasets of different volumes, MiME consistently outperformed all baseline models. Especially, MiME showed the greatest performance improvement (15% relative gain in PR-AUC over the best baseline) for the smallest dataset, demonstrating its ability to effectively model the multilevel structure of EHR data.

## 2  Method

EHR data can be represented by a common hierarchy that begins with individual patient records, where each patient record consists of a sequence of visits. In a typical visit, a physician gives a diagnosis to a patient and then order medications or procedures based on the diagnosis. This process generates a set of treatment (medication and procedure) codes and a relationship among diagnosis and treatment codes (see Figure 1). MiME is designed to explicitly capture the relationship between the diagnosis codes and the treatment codes within visits.

### 2.1  Notations of MiME

Assume a patient has a sequence of visits $\mathcal{V}^{(1)}, \ldots, \mathcal{V}^{(t)}$ over time, where each visit $\mathcal{V}^{(t)}$ contains a varying number of diagnosis (Dx) objects $\mathcal{O}_1^{(t)}, \ldots, \mathcal{O}_{|\mathcal{V}^{(t)}|}^{(t)}$. Each $\mathcal{O}_i^{(t)}$ consists of a single Dx code $d_i^{(t)} \in \mathcal{A}$ and a set of associated treatments (medications or procedures) $\mathcal{M}_i^{(t)}$. Similarly, each $\mathcal{M}_i^{(t)}$ consists of varying number of treatment codes $m_{i,1}^{(t)}, \ldots, m_{i,|\mathcal{M}_i^{(t)}|}^{(t)} \in \mathcal{B}$. To reduce clutter, we omit

Table 1: Notations for `MiME`. Note that the dimension size $z$ is used in many places due to the use of skip-connections, which will be described in section 2.2.

| Notation | Definition |
|---|---|
| $\mathcal{A}$ | Set of unique diagnosis codes |
| $\mathcal{B}$ | Set of unique treatment codes (medications and procedures) |
| $\mathbf{h}$ | A vector representation of a patient |
| $\mathcal{V}^{(t)}$ | A patient's $t$-th visit, which contains diagnosis objects $\mathcal{O}_1^{(t)}, \ldots, \mathcal{O}_{|\mathcal{V}^{(t)}|}^{(t)}$ |
| $\mathbf{v}^{(t)} \in \mathbb{R}^z$ | A vector representation of $\mathcal{V}^{(t)}$ |
| $\mathcal{O}_i^{(t)}$ | $i$-th diagnosis object of $t$-th visit consisting of Dx code $d_i^{(t)}$ and treatment codes $\mathcal{M}_i^{(t)}$ |
| $\mathbf{o}_i^{(t)} \in \mathbb{R}^z$ | A vector representation of $\mathcal{O}_i^{(t)}$ |
| $p(d_i^{(t)}|\mathbf{o}_i^{(t)}), p(m_{i,j}^{(t)}|\mathbf{o}_i^{(t)})$ | Auxiliary predictions, respectively for a Dx code and a treatment code based on $\mathbf{o}_i^{(t)}$ |
| $d_i^{(t)} \in \mathcal{A}$ | Dx code of diagnosis object $\mathcal{O}_i^{(t)}$ |
| $\mathcal{M}_i^{(t)}$ | a set of treatment codes associated with $i$-th Dx code $d_i^{(t)}$ in visit $t$ |
| $m_{i,j}^{(t)} \in \mathcal{B}$ | $j$-th treatment code of $\mathcal{M}_i^{(t)}$ |
| $g(d_i^{(t)}, m_{i,j}^{(t)})$ | A function that captures the interaction between $d_i^{(t)}$ and $m_{i,j}^{(t)}$ |
| $f(d_i^{(t)}, \mathcal{M}_i^{(t)})$ | A function that computes embedding of diagnosis object $\mathbf{o}_i^{(t)}$ |
| $r(\cdot) \in \mathbb{R}^z$ | A helper notation for extracting $d_i^{(t)}$ or $m_{i,j}^{(t)}$'s embedding vector |

the superscript $(t)$ indicating $t$-th visit, when we are discussing a single visit. Table 1 summarizes notations we will use throughout the paper.

In Figure 1, there are five Dx codes, hence five Dx objects $\mathcal{O}_1^{(t)}, \ldots, \mathcal{O}_5^{(t)}$. More specifically, the first Dx object $\mathcal{O}_1$ has $d_1^{(t)} = $ *Fatigue* as the Dx code, but no treatment codes. $\mathcal{O}_2$, on the other hand, has Dx code $d_2^{(t)} = $ *Cough* and two associated treatment codes $m_{2,1}^{(t)} = $ *Benzonatate* and $m_{2,2}^{(t)} = $ *Acetaminophen*. In this case, we can use $g(d_2^{(t)}, m_{2,1}^{(t)})$ to capture the interaction between Dx code *Cough* and treatment code *Benzonatate*, which will be fed to $f(d_2^{(t)}, \mathcal{M}_2^{(t)})$ to obtain the vector representation of Dx object $\mathbf{o}_2^{(t)}$. Using the five Dx object embeddings $\mathbf{o}_1^{(t)}, \ldots, \mathbf{o}_5^{(t)}$, we can obtain a visit embedding $\mathbf{v}^{(t)}$. In addition, some treatment codes (*e.g. Acetaminophen*) can be shared by two or more Dx codes (*e.g. Cough, Fever*), if the doctor ordered a single medication for more than one diagnosis. Then each Dx object will have its own copy of the treatment code attached to it, in this case denoted, $m_{2,2}^{(t)}$ and $m_{3,1}^{(t)}$, respectively.

## 2.2 Description of `MiME`

**Multilevel Embedding** As discussed earlier, previous approaches often flatten a single visit such that Dx codes and treatment codes are packed together so that a single visit $\mathcal{V}^{(t)}$ can be expressed as a binary vector $\mathbf{x}^{(t)} \in \{0, 1\}^{|\mathcal{A}|+|\mathcal{B}|}$ where each dimension corresponds to a specific Dx and treatment code. Then a patient's visit sequence is encoded as:

$$\mathbf{v}^{(t)} = \sigma(\mathbf{W}_x \mathbf{x}^{(t)} + \mathbf{b}_x)$$
$$\mathbf{h} = h(\mathbf{v}^{(1)}, \mathbf{v}^{(2)}, \ldots, \mathbf{v}^{(t)})$$

where $\mathbf{W}_x$ is the embedding matrix that converts the binary vector $\mathbf{x}$ to a lower-dimensional visit representation[3], $\sigma$ a non-linear activation function such as sigmoid or rectified linear unit (ReLU), $h(\cdot)$ a function that maps a sequence of visit representations $\mathbf{v}^{(0)}, \ldots, \mathbf{v}^{(t)}$ to a patient representation $\mathbf{h}$. In contrast, `MiME` effectively derives a visit representation $\mathbf{v}^{(t)}$, than can be plugged into any $h(\cdot)$ for the downstream prediction task. $h(\cdot)$ can simply be an RNN or a combination of RNNs and CNN and attention mechanisms [1].

`MiME` explicitly captures the hierarchy between Dx codes and treatment codes depicted in Figure 1. Figure 2 illustrates how `MiME` builds the representation of $\mathcal{V}$ (omitting the superscript $(t)$) in a bottom-up fashion via multilevel embedding. In a single Dx object $\mathcal{O}_i$, a Dx code $d_i$ and its associated treatment codes $\mathcal{M}_i$ are used to obtain a vector representation of $\mathcal{O}_i$, $\mathbf{o}_i$. Then multiple Dx object embeddings $\mathbf{o}_0, \ldots, \mathbf{o}_{|\mathcal{V}|}$ in a single visit are used to obtain a visit embedding $\mathbf{v}$, which in turn forms

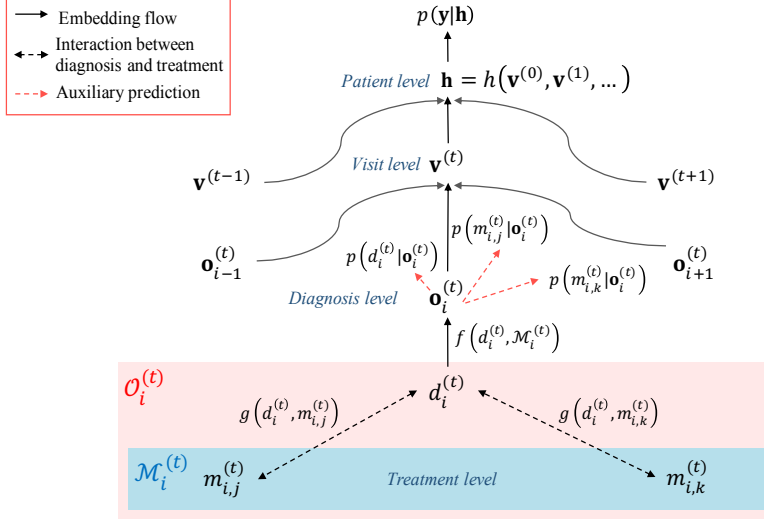

Figure 2: Prediction model using `MiME`. Codes are embedded into multiple levels: diagnosis-level, visit-level, and patient-level. Final prediction $p(\mathbf{y}|\mathbf{h})$ is based on the patient representation $\mathbf{h}$, which is derived from visit representations $\mathbf{v}^{(0)}, \mathbf{v}^{(1)}, \ldots$, where each $\mathbf{v}^{(t)}$ is generated using `MiME` framework. As shown in the *Treatment level*, `MiME` explicitly captures the interactions between a diagnosis code and the associated treatment codes. `MiME` also uses those codes as auxiliary prediction targets to improve generalizability when large training data are not available.

a patient embedding $\mathbf{h}$ with other visit embeddings. The formulation of `MiME` is as follows:

$$\mathbf{v} = \sigma\bigg(\mathbf{W}_v\Big(\underbrace{\sum_i^{|\mathcal{V}|} f(d_i, \mathcal{M}_i)}_{\text{F: used for skip-connection}}\Big)\bigg) + F \tag{1}$$

$$f(d_i, \mathcal{M}_i) = \mathbf{o}_i = \sigma\bigg(\mathbf{W}_o\Big(\underbrace{r(d_i) + \sum_j^{|\mathcal{M}_i|} g(d_i, m_{i,j})}_{\text{G: used for skip-connection}}\Big)\bigg) + G \tag{2}$$

$$g(d_i, m_{i,j}) = \sigma\big(\mathbf{W}_m r(d_i)\big) \odot r(m_{i,j}) \tag{3}$$

where Eq. (1), Eq. (2) and Eq. (3) describe `MiME` in a top-down fashion, respectively corresponding to *Visit level*, *Diagnosis level* and *Treatment level* in Figure 2.

In Eq. (1), a visit embedding $\mathbf{v}$ is obtained by summing Dx object embeddings $\mathbf{o}_1, \ldots, \mathbf{o}_{|\mathcal{V}|}$, which are then transformed with $\mathbf{W}_v \in \mathbb{R}^{z \times z}$. $\sigma$ is a non-linear activation function such as sigmoid or rectified linear unit (ReLU). In Eq. (2), $\mathbf{o}_i$ is obtained by summing $r(d_i) \in \mathbb{R}^z$, the vector representation of the Dx code $d_i$, and the effect of the interactions between $d_i$ and its associated treatments $\mathcal{M}_i$, which are then transformed with $\mathbf{W}_o \in \mathbb{R}^{z \times z}$. The interactions captured by $g(d_i, m_{i,j})$ are added to the $r(d_i)$, which can be interpreted as adjusting the diagnosis representation according to its associated treatments (medications and procedures). Note that in both Eq. (1) and Eq. (2), $F$ and $G$ are used to denote skip-connections [23].

In Eq. (3), the interaction between a Dx code embedding $r(d_i)$ and a treatment code embedding $r(m_{i,j})$ is captured by element-wise multiplication $\odot$. Weight matrix $\mathbf{W}_m \in \mathbb{R}^{z \times z}$ sends the Dx code embedding $r(d_i)$ into another latent space, where the interaction between $d_i$ and the corresponding $m_{i,j}$ can be effectively captured. The formulation of Eq. (3) was inspired by recent developments in bilinear pooling technique [37, 21, 19, 24], which we discuss in more detail in Appendix A. With Eq. (3) in mind, $G$ in Eq. (2) can also be interpreted as $r(d_i)$ being skip-connected to the sum of interactions $g(d_i, m_{i,j})$.

**Joint Training with Auxiliary Tasks** Patient embedding $\mathbf{h}$ is often used for specific prediction tasks, such as heart failure prediction or mortality. The representation power of $\mathbf{h}$ comes from properly capturing each visit $\mathcal{V}^{(t)}$, and modeling the longitudinal aspect with the function $h(\mathbf{v}_0, \ldots, \mathbf{v}_t)$. Since the focus of this work is on modeling a single visit $\mathcal{V}^{(t)}$, we perform auxiliary predictions as follows:

$$\hat{d}_i^{(t)} = p(d_i^{(t)}|\mathbf{o}_i^{(t)}) = \text{softmax}(\mathbf{U}_d \mathbf{o}_i^{(t)}) \tag{4}$$

$$\hat{m}_{i,j}^{(t)} = p(m_{i,j}^{(t)}|\mathbf{o}_i^{(t)}) = \sigma(\mathbf{U}_m \mathbf{o}_i^{(t)}) \tag{5}$$

$$L_{aux} = -\lambda_{aux} \sum_t^T \left( \sum_i^{|\mathcal{V}^{(t)}|} \left( CE(d_i^{(t)}, \hat{d}_i^{(t)}) + \sum_j^{|\mathcal{M}_i^{(t)}|} CE(m_{i,j}^{(t)}, \hat{m}_{i,j}^{(t)}) \right) \right) \tag{6}$$

Given Dx object embeddings $\mathbf{o}_1^{(t)}, \ldots, \mathbf{o}_{|\mathcal{V}^{(t)}|}^{(t)}$, while aggregating them to obtain $\mathbf{v}^{(t)}$ as in Eq. (1), MiME predicts the Dx code $d_i^{(t)}$, and the associated treatment code $m_{i,j}^{(t)}$ as depicted by Figure 2. In Eq. (4) and Eq. (5), $\mathbf{U}_d \in \mathbb{R}^{|\mathcal{A}| \times z}$ and $\mathbf{U}_m \in \mathbb{R}^{|\mathcal{B}| \times z}$ are weight matrices used to compute the the prediction of Dx code $\hat{d}_i^{(t)}$ and the prediction of the treatment code $\hat{m}_{i,j}^{(t)}$, respectively. In Eq. (6), $T$ denotes the total number of visits the patient made, $CE(\cdot, \cdot)$ the cross-entropy function and $\lambda_{aux}$ the coefficient for the auxiliary loss term. We used the softmax function for predicting $d_i^{(t)}$ since in a single Dx object $\mathcal{O}_i^{(t)}$, there is only one Dx code involved. However, there could be no (or many) treatment codes associated with $\mathcal{O}_i^{(t)}$, and therefore we used $|\mathcal{B}|$ number of sigmoid functions for predicting each treatment code.

Auxiliary tasks are based on the inherent structure of the EHR data, and require no additional labeling effort. These auxiliary tasks guide the model to learn Dx object embeddings $\mathbf{o}_i^{(t)}$ that are representative of the specific codes involved with it. Correctly capturing the events within a visit is the basis of all downstream prediction tasks, and these general-purpose auxiliary tasks, combined with the specific target task, encourage the model to learn visit embeddings $\mathbf{v}^{(t)}$ that are not only tuned for the target prediction task, but also grounded in general-purpose foundational knowledge.

## 3 Experiments

In this section, we first describe the dataset and the baseline models, and present evaluation results. The source code of MiME is publicly available at `https://github.com/mp2893/mime`.

### 3.1 Source of Data

We conducted all our experiments using EHR data provided by Sutter Health. The dataset was constructed for a study designed to predict a future diagnosis of heart failure, and included EHR data from 30,764 senior patients 50 to 85 years of age. We extracted the diagnosis codes, medication codes and the procedure codes from encounter records, and related orders. We used Clinical Classification Software for ICD9-CM[4] to group the ICD9 diagnosis codes into 388 categories. Generic Product Identifier Drug Group[5] was used to group the medication codes into 99 categories. Clinical Classifications Software for Services and Procedures[6] was used to group the CPT procedure codes into 1,824 categories. Any code that did not fit into the grouper formed its own category. Table 2 summarizes data statistics.

### 3.2 Baseline Models

First, we use Gated Recurrent Units (GRU) [9] with different embedding strategies to map visit embedding sequence $\mathbf{v}^{(1)}, \ldots, \mathbf{v}^{(T)}$ to a patient representation $\mathbf{h}$:

- **raw**: A single visit $\mathcal{V}^{(t)}$ is represented by a binary vector $\mathbf{x}^{(t)} \in \{0,1\}^{|\mathcal{A}|+|\mathcal{B}|}$. Only the dimensions corresponding to the codes occurring in that visit is set to 1, and the rest are 0.

Table 2: Statistics of the dataset

| # of patients | 30,764 |
|---|---|
| # of visits | 616,073 |
| Avg. # of visits per patient | 20.0 |
| # of unique codes | 2,311 (Dx:388, Rx:99, Proc:1,824) |
| Avg. # of Dx per visit | 1.93 (Max: 29) |
| Avg. # of Rx per diagnosis | 0.31 (Max: 17) |
| Avg. # of Proc. per diagnosis | 0.36 (Max: 10) |

- **linear**: The binary vector $\mathbf{x}^{(t)}$ is linearly transformed to a lower-dimensional vector $\mathbf{v}^{(t)} = \mathbf{W}_x\mathbf{x}^{(t)}$ where $\mathbf{W}_x \in \mathbb{R}^{b \times (|\mathcal{A}|+|\mathcal{B}|)}$ is the embedding matrix. This is equivalent to taking the vector representations of the codes (*i.e.* columns of the embedding matrix $\mathbf{W}_x$) in the visit $\mathcal{V}^{(t)}$, and summing them up to derive a single vector $\mathbf{v}^{(t)} \in \mathbb{R}^b$.

- **sigmoid, tanh, relu**: The binary vector $\mathbf{x}^{(t)}$ is transformed to a lower-dimensional vector $\mathbf{v}^{(t)} = \sigma(\mathbf{W}_x\mathbf{x}^{(t)})$ where we use either *sigmoid*, *tanh*, or *ReLU* for $\sigma(\cdot)$ to add non-linearity to **linear**.

- **sigmoid$_{mlp}$, tanh$_{mlp}$, relu$_{mlp}$**: We add one more layer to **sigmoid**, **tanh** and **relu** to increase their expressivity. The visit embedding is now $\mathbf{v}^{(t)} = \sigma(\mathbf{W}_{x_2}\sigma(\mathbf{W}_{x_1}\mathbf{x}^{(t)}))$ where $\sigma$ is either sigmoid, tanh or ReLU. We do not test **linear$_{mlp}$** since two consecutive linear layers can be collapsed to a single linear layer.

Second, we also compare with two advanced embedding methods that are specific designed for modeling EHR data.

- **Med2Vec**: We use Med2Vec [11] to learn visit representations, and use those fixed vectors as input to the prediction model. We test this model as a representative case of unsupervised embedding approach using EHR data.

- **GRAM**: We use GRAM [12], which is equivalent to injecting domain knowledge (ICD9 Dx code tree) to **tanh** via attention mechanism. We test this model as a representative case of incorporating external domain knowledge.

### 3.3 Prediction Tasks

*Heart failure prediction* The objective is to predict the first diagnosis of heart failure (HF), given an 18-months observation records discussed in section 3.1. Among 30,764 patients, 3,414 were case patients who were diagnosed with HF within a 1-year window after the 18-months observation. The remaining 27,350 patients were controls. The case-control selection criteria are detailed in [39] and summarized in Appendix B. While an accurate prediction of HF can save a large amount of costs and lives [33], this task is also suitable for assessing how well a model can learn the relationship between the external label (*i.e.* the label information is not inherent in the EHR data) and the features (*i.e.* codes).

We applied logistic regression to the patient representation $\mathbf{h}$ to obtain a value between 0 (no HF onset) and 1 (HF onset). All models were trained end-to-end except **Med2Vec**. We report Area under the Precision-Recall Curve (PR-AUC) in the experiment and Area under the Receiver Operating Characteristic (ROC-AUC) in the appendix, as PR-AUC is considered a better measure for imbalanced data like ours [34, 16]. Implementation and training configurations are described in Appendix C. We also performed *sequential disease prediction (SDP)* (predicting all diagnoses of the next visit at every timestep) where MiME demonstrated superior performance over all baseline models. The detailed description and results of SDP are provided in Appendix H and Appendix I respectively.

### 3.4 Experiment 1: Varying the Data Size

To evaluate MiME's performance in another perspective, we created four datasets $E_1, E_2, E_3, E_4$ from the original data such that each dataset consisted of patients with varying maximum sequence length $T_{max}$ (*i.e.* maximum number of visits). In order to simulate a new hospital collecting patient records over time, we increased $T_{max}$ for each dataset such that $10, 20, 30, 150$ for $E_1, E_2, E_3, E_4$ respectively. Each dataset had 6299 (414 cases), 15794 (1177 cases), 21128 (1848 cases), 27428

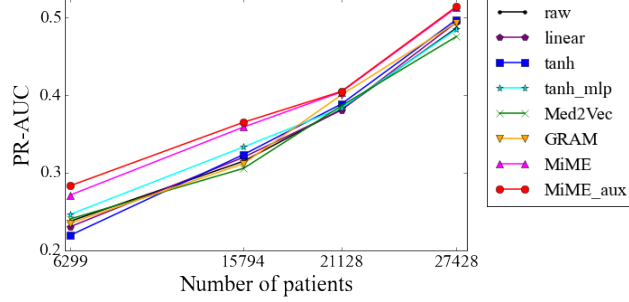

Figure 3: Test PR-AUC of HF prediction for increasing data size. A table with the results of all baseline models is provided in Appendix F

(3173 cases) patients respectively. For MiME $_{aux}$, we used the same $0.015$ for the auxiliary loss coefficient $\lambda_{aux}$.

Figure 3 shows the test PR-AUC for HF prediction across all datasets (loss and ROC-AUC are described in Appendix G). Again we show the strongest activation functions **tanh** and **tanh**$_{mlp}$ here and provide the full table in Appendix F. We can readily see that MiME outperforms all baseline models across all datasets. However, the performance gap between MiME and the baselines are larger in datasets $E_1, E_2$ than in datasets $E_3, E_4$, confirming our assumption that exploiting the inherent structure of EHR can alleviate the data insufficiency problem. Especially for the smallest dataset $E_1$, MiME $_{aux}$ (0.2831 PR-AUC) demonstrated significantly better performance than the best baseline **tanh**$_{mlp}$ (0.2462 PR-AUC), showing $15\%$ relative improvement.

It is notable that MiME consistently outperformed **GRAM** in both Table 3 and Figure 3 in terms of test loss and test PR-AUC. To be fair, GRAM was only using Dx code hierarchy (thus ungrouped 5814 Dx codes were used), and no additional domain knowledge regarding treatment codes. However, the experiment results tell us that even without resorting to external domain knowledge, we can still gain improved predictive performance by carefully studying the EHR data and leveraging its inherent structure.

## 3.5   Experiment 2: Varying Visit Complexity

Table 3: HF prediction performance on small datasets. Values in the parentheses denote standard deviations from 5-fold random data splits. All models used GRU for mapping the visit embeddings $\mathbf{v}^{(1)}, \ldots, \mathbf{v}^{(T)}$ to a patient representation $\mathbf{h}$. Two best values in each column are marked in bold. A full table with all baselines is provided in Appendix D.

| | $\mathbf{D_1}$ (Visit complexity 0-15%) (5608 patients, 464 cases) | | $\mathbf{D_2}$ (Visit complexity 15-30%) (5180 patients, 341 cases) | | $\mathbf{D_3}$ (Visit complexity 30-100%) (5231 patients, 383 cases) | |
|---|---|---|---|---|---|---|
| | test loss | test PR-AUC | test loss | test PR-AUC | test loss | test PR-AUC |
| raw | 0.2553 (0.0084) | 0.2669 (0.0314) | 0.2203 (0.0186) | 0.2388 (0.0460) | 0.2144 (0.0127) | 0.3776 (0.0589) |
| linear | 0.2562 (0.0108) | 0.2722 (0.0354) | 0.2200 (0.0187) | 0.2403 (0.0229) | 0.2021 (0.0176) | 0.4339 (0.0411) |
| tanh | 0.2648 (0.0124) | 0.2707 (0.0138) | 0.2186 (0.0182) | 0.2479 (0.0512) | 0.2025 (0.0151) | 0.4415 (0.0532) |
| tanh$_{mlp}$ | 0.2587 (0.0121) | 0.2671 (0.0257) | 0.2289 (0.0213) | 0.2296 (0.0185) | 0.2024 (0.0181) | 0.4290 (0.0510) |
| Med2Vec | 0.2601 (0.0186) | **0.2771** (0.0288) | 0.2171 (0.0170) | 0.2356 (0.0309) | 0.2044 (0.0129) | 0.3813 (0.0240) |
| GRAM | 0.2554 (0.0254) | 0.2633 (0.0521) | 0.2249 (0.0448) | 0.2505 (0.0609) | 0.2333 (0.0362) | 0.3998 (0.0628) |
| MiME | **0.2535** (0.0042) | 0.2637 (0.0326) | **0.2121** (0.0238) | **0.2579** (0.0241) | **0.1931** (0.0140) | **0.4685** (0.0432) |
| MiME$_{aux}$ | **0.2512** (0.0073) | **0.2750** (0.0326) | **0.2117** (0.0238) | **0.2589** (0.0287) | **0.1910** (0.0163) | **0.4787** (0.0434) |

Next, we conducted a series of experiments to confirm that MiME can indeed capture the relationship between Dx codes and treatment codes, thus producing robust performance in small datasets. Specifically, we created three small datasets $D_1, D_2, D_3$ from the original data such that each dataset consisted of patients with varying degree of Dx-treatment interactions (*i.e.* visit complexity). We defined *visit complexity* as below to calculate for a patient the percentage of visits that have at least two diagnosis codes associated with different sets of treatment codes,

$$\text{visit complexity} = \frac{\#\mathcal{V}^{(t)} \text{ where } |set(\mathcal{M}_1^{(t)}, \ldots, \mathcal{M}_{|\mathcal{V}^{(t)}|}^{(t)})| \geq 2}{T}$$

where $T$ denotes the total number of visits. For example, in Figure 1, the $t$-th visit $\mathcal{V}^{(t)}$ has *Fever* associated with no treatments, and *Cough* associated with two treatments. Therefore $\mathcal{V}^{(t)}$ qualifies as a complex visit. From the original dataset, we selected patients with a short sequence (less than 20 visits) to simulate a hospital newly equipped with a EHR system, and there aren't much data collected yet. Among the patients with less than 20 visits, we used visit complexity ranges $0 - 15\%, 15 - 30\%, 30 - 100\%$ to create $D_1, D_2, D_3$ consisting of 5608 (464 HF cases), 5180 (341 HF cases), 5231 (383 HF cases) patients respectively. For training MiME with auxiliary tasks, we explored various $\lambda_{aux}$ values between $0.01 - 0.1$, and found $0.015$ to provide the best performance, although other values also improved the performance in varying degrees.

Table 3 shows the HF prediction performance for the dataset $D_1, D_2$ and $D_3$. To enhance readability, we show here the results of the strongest activation function **tanh** and **tanh**$_{mlp}$, and we report test loss and test PR-AUC. The results of other activation functions and the test ROC-AUC are provided in Appendix D and Appendix E.

Table 3 provides two important messages. First of all, both MiME and MiME $_{aux}$ show close to the best performance in all datasets $D_1, D_2$ and $D_3$, especially high complexity dataset $D_3$. This confirms that MiME indeed draws its power from the interactions between Dx codes and treatment codes, with or without the auxiliary tasks. In $D_1$, patients' visits do not have much structure, that it makes little difference whether we use MiME or not, and its performance is more or less similar to many baselines. Second, auxiliary tasks indeed help MiME generalize better to patients unseen during training. In all datasets $D_1, D_2$ and $D_3$, MiME $_{aux}$ outperforms MiME in all measures, especially in $D_3$ where it shows PR-AUC 0.4787 (8.4% relative improvement over the best baseline **tanh**).

# 4 Related Work

Over the years, medical concept embedding has been an active research area. Some works tried to summarize sparse and high-dimensional medical concepts into compressed vectors [15, 18]. In those works, medical concepts were organized as temporal sequences, from which embeddings were derived. Other works used latent layers of deep models for representing more abstract medical concepts [14, 10, 13, 12, 27, 2]. For example, restricted Boltzmann Machines, stacked auto-encoders or multi-layer neural networks were used to learn the representation of codes, visits, or patients [38, 28, 11]. Some works used medical ontologies to learn medical concept representations [12, 8]. Although all works successfully learned concept embeddings for some task in varying degrees, they did not fully utilize the multilevel structure or diagnosis-treatment relationship of EHR.

Recently, multiple code types in EHR gained more attentions. In [35], authors viewed different code types separately, and tried to capture complex relationships across these disparate data types using RNNs, but they did not explicitly address the hierarchy of EHR data. More recently in [30], the authors tried to explicitly capture the interaction between a set of all diagnosis codes and a set of all medication codes occurring in a visit. However, in their experiment, simply concatenating both sets to obtain a visit vector outperformed other methods in many tasks. This suggests that disregarding the diagnosis-specific Dx-Rx interaction and flattening all codes as sets is a suboptimal approach to modeling EHR data.

As described in section 2.2, we employ auxiliary task strategy to train a robust model. Training a model to predict multiple related targets has shown to improve model robustness in medical prediction tasks in previous studies. For example, [5] used lab values as auxiliary targets to improve mortality prediction performance. More recent studies [29, 22, 4] demonstrated improved prediction accuracy when training a model with multiple related tasks such as mortality prediction and phenotyping.

# 5 Conclusion

In this work, we presented MiME, an integrated approach that simultaneously models hierarchical inter-code relations into medical concept embedding while jointly performing auxiliary prediction tasks. Through extensive empirical evaluation, MiME demonstrated impressive performance across all benchmark tasks and its generalization ability to smaller datasets, especially outperforming baselines in terms of PR-AUC in heart failure prediction. As we have established in this work that MiME can be a good choice for modeling visits, in the future, we plan to extend MiME to include more fine-grained medical events such as procedure outcomes, demographic information, and medication instructions.

**Acknowledgments**

This work was supported by the National Science Foundation, award IIS-#1418511 and CCF-#1533768, the National Institute of Health award 1R01MD011682-01 and R56HL138415, and Samsung Scholarship. We would also like to thank Sherry Yan for her helpful comments on the original manuscript.

## Footnotes

[3]We omit bias variables throughout the paper to reduce clutter.

[4]https://www.hcup-us.ahrq.gov/toolssoftware/ccs/ccs.jsp

[5]http://www.wolterskluwercdi.com/drug-data/medi-span-electronic-drug-file/

[6]https://www.hcup-us.ahrq.gov/toolssoftware /ccs_svcsproc/ccssvcproc.jsp

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
