[Supplementary Material]

Table 4: Qualifying ICD-9 codes for heart failure

| ICD-9 Code | Description |
| --- | --- |
| 398.91 | Rheumatic heart failure (congestive) |
| 402.01 | Malignant hypertensive heart disease with heart failure |
| 402.11 | Benign hypertensive heart disease with heart failure |
| 402.91 | Unspecified hypertensive heart disease with heart failure |
| 404.01 | Hypertensive heart and chronic kidney disease, malignant, with heart failure and with chronic kidney disease stage I through stage IV, or unspecified |
| 404.03 | Hypertensive heart and chronic kidney disease, malignant, with heart failure and with chronic kidney disease stage V or end stage renal disease |
| 404.11 | Hypertensive heart and chronic kidney disease, benign, with heart failure and with chronic kidney disease stage I through stage IV, or unspecified |
| 404.13 | Hypertensive heart and chronic kidney disease, benign, with heart failure and chronic kidney disease stage V or end stage renal disease |
| 404.91 | Hypertensive heart and chronic kidney disease, unspecified, with heart failure and with chronic kidney disease stage I through stage IV, or unspecified |
| 404.93 | Hypertensive heart and chronic kidney disease, unspecified, with heart failure and chronic kidney disease stage V or end stage renal disease |
| 428.0 | Congestive heart failure, unspecified |
| 428.1 | Left heart failure |
| 428.20 | Systolic heart failure, unspecified |
| 428.21 | Acute systolic heart failure |
| 428.22 | Chronic systolic heart failure |
| 428.23 | Acute on chronic systolic heart failure |
| 428.30 | Diastolic heart failure, unspecified |
| 428.31 | Acute diastolic heart failure |
| 428.32 | Chronic diastolic heart failure |
| 428.33 | Acute on chronic diastolic heart failure |
| 428.40 | Combined systolic and diastolic heart failure, unspecified |
| 428.41 | Acute combined systolic and diastolic heart failure |
| 428.42 | Chronic combined systolic and diastolic heart failure |
| 428.43 | Acute on chronic combined systolic and diastolic heart failure |
| 428.9 | Heart failure, unspecified |

# A    Discussion of Bilinear Pooling

In Eq. (3), $g(d_i, m_{i,j})$ uses a form of bilinear pooling to explicitly capture the interaction between the Dx code and the treatment code. The original bilinear pooling [37] derives a scalar feature $f_i$ between two embeddings $\mathbf{x}, \mathbf{y}$ such that $f_i = \mathbf{x}^T \mathbf{W}_i \mathbf{y}$ where $\mathbf{W}_i$ is a trainable weight matrix. Since we typically extract many features $f_0, \ldots, f_i$, to capture the interaction between two embeddings, bilinear pooling requires us to train multiple weight matrices (*i.e.* weight tensor). Due to this requirement, researchers developed more efficient methods such as compact bilinear pooling [21, 19] and low-rank bilinear pooling [24], which is used in this work.

# B    Heart Failure Case-Control Selection Criteria

Case patients were 40 to 85 years of age at the time of HF diagnosis. HF diagnosis (HFDx) is defined as: 1) Qualifying ICD-9 codes for HF appeared in the encounter records or medication orders. Qualifying ICD-9 codes are displayed in Table 4. 2) a minimum of three clinical encounters with qualifying ICD-9 codes had to occur within 12 months of each other, where the date of diagnosis was assigned to the earliest of the three dates. If the time span between the first and second appearances of the HF diagnostic code was greater than 12 months, the date of the second encounter was used as the first qualifying encounter. The date at which HF diagnosis was given to the case is denoted as HFDx. Up to ten eligible controls (in terms of sex, age, location) were selected for each case, yielding an overall ratio of 9 controls per case. Each control was also assigned an index date, which is the HFDx of the matched case. Controls are selected such that they did not meet the operational criteria for HF diagnosis prior to the HFDx plus 182 days of their corresponding case. Control subjects were required to have their first office encounter within one year of the matching HF case patient's first

Table 5: HF prediction performance of all models on small datasets. Values in the parentheses denote standard deviations from 5-fold random data splits. Two best values in each column are marked in bold.

| | $D_1$ | | $D_2$ | | $D_3$ | |
| | (Visit complexity 0-15%, 5608 patients) | | (Visit complexity 15-30%, 5180 patients) | | (Visit complexity 30-100%, 5231 patients) | |
| | test loss | test PR-AUC | test loss | test PR-AUC | test loss | test PR-AUC |
|---|---|---|---|---|---|---|
| raw | 0.2553 (0.0084) | 0.2669 (0.0314) | 0.2203 (0.0186) | 0.2388 (0.0460) | 0.2144 (0.0127) | 0.3776 (0.0589) |
| linear | 0.2562 (0.0108) | 0.2722 (0.0354) | 0.2200 (0.0187) | 0.2403 (0.0229) | 0.2021 (0.0176) | 0.4339 (0.0411) |
| sigmoid | 0.2594 (0.0062) | 0.2637 (0.0374) | 0.2198 (0.0220) | 0.2445 (0.0363) | 0.2029 (0.0118) | 0.4358 (0.0585) |
| tanh | 0.2648 (0.0124) | 0.2707 (0.0138) | 0.2186 (0.0182) | 0.2479 (0.0512) | 0.2025 (0.0151) | 0.4415 (0.0532) |
| relu | 0.2601 (0.0107) | 0.2546 (0.0109) | 0.2288 (0.0244) | 0.1957 (0.0217) | 0.2083 (0.0124) | 0.4100 (0.0276) |
| $sigmoid_{mlp}$ | 0.2836 (0.0102) | 0.1207 (0.0145) | 0.2407 (0.0162) | 0.1119 (0.0334) | 0.2127 (0.0294) | 0.3547 (0.1208) |
| $tanh_{mlp}$ | 0.2587 (0.0121) | 0.2671 (0.0257) | 0.2289 (0.0213) | 0.2296 (0.0185) | 0.2024 (0.0181) | 0.4290 (0.0510) |
| $relu_{mlp}$ | 0.2650 (0.0088) | 0.2463 (0.0148) | 0.2288 (0.0235) | 0.1982 (0.0298) | 0.2144 (0.0202) | 0.3872 (0.0476) |
| Med2Vec | 0.2601 (0.0186) | **0.2771** (0.0288) | 0.2171 (0.0170) | 0.2356 (0.0309) | 0.2044 (0.0129) | 0.3813 (0.0240) |
| GRAM | 0.2554 (0.0254) | 0.2633 (0.0521) | 0.2249 (0.0448) | 0.2505 (0.0609) | 0.2333 (0.0362) | 0.3998 (0.0628) |
| MiME | **0.2535** (0.0042) | 0.2637 (0.0326) | **0.2121** (0.0238) | **0.2579** (0.0241) | **0.1931** (0.0140) | **0.4685** (0.0432) |
| MiME $_{aux}$ | **0.2512** (0.0073) | **0.2750** (0.0326) | **0.2117** (0.0238) | **0.2589** (0.0287) | **0.1910** (0.0163) | **0.4787** (0.0434) |

office visit, and have at least one office encounter 30 days before or any time after the case's HF diagnosis date to ensure similar duration of observations among cases and controls.

## C  Training Details

All models were implemented in TensorFlow 1.4 [36], and trained with a system equipped with Intel Xeon E5-2620, 512TB memories and 8 Nvidia Pascal Titan X's. We used Adam [25] for optimization, with the learning rate $1e-3$.

In all experiments, the reported results are averaged over 5-fold random data splits: training (70%), validation (10%) and test (20%). All models were trained with the minibatch of 20 patients for 20,000 iterations to guarantee convergence. At every 100 iterations, we evaluated the loss value of the validation set for early stopping.

For the non-linear activation functions in MiME, we used ReLU in all places except for the one in Eq. (1) where we used sigmoid to benefit from its regularization effect. We avoid the vanishing gradient problem by using the skip connections. Note that simply adding skip connections to **sigmoid**$_{mlp}$ did not improve performance.

For the first experiment in section 3.5, size of the visit vector **v** was 128 in all baseline models except **raw**. We ran a number of preliminary experiments with values 64, 128, 256 and 512, and we concluded that 128 was sufficient for all models to obtain optimal performance, as the datasets $D_1, D_2$ and $D_3$ were rather small. For MiME, we adjusted the size of the embeddings $z$ to match the number of parameters to the baselines. **Med2Vec** was also trained to obtain 128 dimensional visit vectors. Note that **sigmoid**$_{mlp}$, **tanh**$_{mlp}$, **relu**$_{mlp}$ and **GRAM** used $128 \times 128$ more parameters than other models. We used $L_2$ regularization with the coefficient $1e-4$ for all models. We did not use any dropout technique. All models used GRU for the function $h(\mathbf{v}^{(1)}, \ldots, \mathbf{v}^{(T)})$ as described in section 3.3, the cell size of which was 128.

For the second experiment in section 3.4, where the models were trained on gradually larger datasets $E_1, E_2, E_3$ and $E_4$, the size of **v** was set to 256 for all baseline models except **raw**. The same adjustments were made to MiME as before, and the cell size of GRU was also set to 256.

## D  Heart Failure Prediction Performance on Datasets $D_1, D_2$ and $D_3$, Full Version

Table 5 shows the performance of all models on datasets $D_1, D_2$ and $D_3$. An interesting finding is that both **sigmoid** and **tanh** mostly outperform **relu** in both measures in $D_1, D_2$ and $D_3$, although *ReLU* is the preferred nonlinear activation for hidden layers in many studies . This seems due to the regularizing effect of *sigmoid* and *tanh* functions. Whereas *ReLU* can produce outputs as high as infinity, *sigmoid* and *tanh* have bounded outputs. Considering that **sigmoid**, **tanh** and **relu** all sum up the code embeddings in a visit $\mathcal{V}^{(t)}$ before applying the nonlinear activation, constraining the output of the nonlinear activation seems to work favorably, especially in $D_3$ where there are more

codes per visit. This regularization benefit, however, diminishes as the dataset grows, which can be confirmed by Table 7 in section F. In addition, as can be seen by the performance of **sigmoid**$_{mlp}$, *sigmoid* clearly suffers from the vanishing gradient problem as opposed to *tanh* or *ReLU* that have larger gradient values.

## E  ROC-AUC of Heart Failure Prediction on Datasets $D_1$, $D_2$ and $D_3$

Table 6: ROC-AUC of all models for HF prediction on small datasets. Values in the parentheses denote standard deviations from 5-fold random data splits. Two best values in each column are marked in bold.

| | **$D_1$** <br> (Visit complexity 0-15%, 5608 patients) | **$D_2$** <br> (Visit complexity 15-30%, 5180 patients) | **$D_3$** <br> (Visit complexity 30-100%, 5231 patients) |
|---|---|---|---|
| raw | 0.7424 (0.0153) | 0.7508 (0.0254) | 0.8130 (0.0315) |
| linear | 0.7298 (0.0187) | 0.7241 (0.0220) | 0.8209 (0.0130) |
| sigmoid | 0.7220 (0.0098) | 0.7331 (0.0475) | 0.8280 (0.0128) |
| tanh | 0.7273 (0.0050) | 0.7244 (0.0175) | 0.8171 (0.0151) |
| relu | 0.7326 (0.0133) | 0.7078 (0.0181) | 0.8166 (0.0211) |
| sigmoid$_{mlp}$ | 0.5520 (0.0136) | 0.5770 (0.0416) | 0.7718 (0.0826) |
| tanh$_{mlp}$ | 0.7215 (0.0188) | 0.7058 (0.0261) | 0.8080 (0.0258) |
| relu$_{mlp}$ | 0.7205 (0.0122) | 0.7014 (0.0177) | 0.7993 (0.0212) |
| Med2Vec | 0.7447 (0.0194) | 0.7515 (0.0243) | 0.8325 (0.0254) |
| GRAM | **0.7586** (0.0240) | 0.6930 (0.0379) | 0.7785 (0.0260) |
| MiME | 0.7433 (0.0127) | **0.7723** (0.0232) | **0.8393** (0.0281) |
| MiME $_{aux}$ | **0.7449** (0.0117) | **0.7741** (0.0209) | **0.8437** (0.0244) |

Table 6 shows ROC-AUC of all models on datasets $D_1$, $D_2$ and $D_3$. Except for $D_1$ where patients have low visit complexity, MiME again consistently outperforms all baseline models. However, the ROC-AUC gap between MiME and baselines is not as great as PR-AUC. This is because ROC-AUC is determined by sensitivity (*i.e.* recall, or true positive rate) and specificity (*i.e.* true negative rate). A model achieves a high specificity if it can correctly identify as many negative samples as possible, which is easier for problems with many negative samples and few positive samples. PR-AUC, on the other hand, is determined by precision and recall. Therefore, for a model to achieve a high PR-AUC, it must correctly retrieve as many positive samples as possible while ignoring negative samples, which is harder for problems with few positive samples.

For heart failure (HF) prediction, achieving high specificity is relatively easy as there are way more controls (*i.e.* negative samples) than cases (*i.e.* positive samples). However, correctly identifying cases while ignoring controls requires a model to recognize what differentiates cases from controls. This means paying attention to the details of the patient records, such as the relationship between the diagnosis codes and treatment codes. That is why MiME shows significant improvement in PR-AUC while showing moderate improvement in ROC-AUC. Also, this also explains why **Med2Vec** shows very poor PR-AUC as opposed to its competitive ROC-AUC. Med2Vec only pays attention to the co-occurrence of codes within a single visit, and not the interaction between diagnosis codes and treatment codes. It can work as a very efficient code grouper (codes that often appear in the same visit end up having similar code embeddings), leading to a increased ROC-AUC. But it cannot achieve a high PR-AUC, as that code grouping loses much of the subtle interaction between diagnosis codes and medication codes.

## F  Test PR-AUC on Datasets $E_1$, $E_2$, $E_3$ and $E_4$, Full Version

Table 7 shows the PR-AUC of all models on datasets $E_1$, $E_2$, $E_3$ and $E_4$. It is notable that some baseline models show fluctuating performance as dataset grows. For example, **tanh**$_{mlp}$ showed competitive performance in small datasets, but weaker performance in large datasets. **relu**$_{mlp}$, on the other hand, did not stand out in small datasets, but became the best baseline in large datasets. Such behaviors, along with the finding in Appendix D regarding the regularization effect, suggest that we should carefully choose activation functions of our model depending on the dataset size.

## G  Test Loss and Test ROC-AUC on Datasets $E_1$, $E_2$, $E_3$ and $E_4$

Table 8 and Table 9 respectively shows the test loss and test ROC-AUC of all models on datasets of varying sizes $E_1$, $E_2$, $E_3$ and $E_4$. Both MiME and MiME $_{aux}$ consistently outperformed all baselines

Table 7: Test PR-AUC of HF prediction for increasing data size. Parentheses denote standard deviations from 5-fold random data splits. The two strongest values in each column are marked bold.

| | $E_1$ (6299 patients) | $E_2$ (15794 patients) | $E_3$ (21128 patients) | $E_4$ (27428 patients) |
|---|---|---|---|---|
| raw | 0.2374 (0.0514) | 0.3149 (0.0367) | 0.3816 (0.0290) | 0.4865 (0.0219) |
| linear | 0.2303 (0.0467) | 0.3200 (0.0353) | 0.3806 (0.0271) | 0.4939 (0.0159) |
| sigmoid | 0.2354 (0.0355) | 0.3260 (0.0392) | 0.3851 (0.0235) | 0.4823 (0.0195) |
| tanh | 0.2192 (0.0407) | 0.3235 (0.0441) | 0.3884 (0.0310) | 0.4973 (0.0262) |
| relu | 0.2293 (0.0459) | 0.3274 (0.0359) | 0.3793 (0.0291) | 0.4957 (0.0160) |
| $\text{sigmoid}_{mlp}$ | 0.0843 (0.0154) | 0.0919 (0.0110) | 0.1333 (0.0047) | 0.2221 (0.0146) |
| $\text{tanh}_{mlp}$ | 0.2462 (0.0675) | 0.3333 (0.0387) | 0.3834 (0.0209) | 0.4847 (0.0172) |
| $\text{relu}_{mlp}$ | 0.2353 (0.0335) | 0.3111 (0.0494) | 0.3976 (0.0235) | 0.4983 (0.0229) |
| Med2Vec | 0.2404 (0.0228) | 0.3057 (0.0508) | 0.3861 (0.0343) | 0.4756 (0.0148) |
| GRAM | 0.2349 (0.0424) | 0.3118 (0.0337) | 0.4002 (0.0113) | 0.4936 (0.0199) |
| MiME | **0.2711** (0.0308) | **0.3589** (0.0533) | **0.4041** (0.0231) | **0.5129** (0.0204) |
| MiME $_{aux}$ | **0.2831** (0.0425) | **0.3651** (0.0473) | **0.4047** (0.0276) | **0.5142** (0.0210) |

in terms of both test loss and test ROC-AUC, except **Med2Vec**. Moreover, MiME $_{aux}$ always showed better performance than MiME except test loss in $E_4$, especially for the smallest dataset $E_1$, confirming our assumption that auxiliary tasks can train a robust model when large datasets are unavailable. $\textbf{tanh}_{mlp}$ consistently showed good performance in terms of ROC-AUC across all datasets, as opposed to showing fluctuating PR-AUC in Table 7. **Med2Vec** again showed a competitive ROC-AUC in all datasets, even outperforming MiME $_{aux}$ in $E_3$. This suggests that initializing MiME's code embeddings with Med2Vec can be an interesting future direction as it may lead to an even better performance.

Table 8: Test loss of HF prediction for increasing data size. Parentheses denote standard deviations from 5-fold random data splits. Two best values in each column are marked bold.

| | $E_1$ (6299 patients) | $E_2$ (15794 patients) | $E_3$ (21128 patients) | $E_4$ (27428 patients) |
|---|---|---|---|---|
| raw | 0.2204 (0.0090) | 0.2236 (0.0166) | 0.2387 (0.0045) | 0.2658 (0.0095) |
| linear | 0.2229 (0.0078) | 0.2245 (0.0160) | 0.2395 (0.0068) | 0.2642 (0.0099) |
| sigmoid | 0.2229 (0.0064) | 0.2215 (0.0135) | 0.2373 (0.0034) | 0.2655 (0.0095) |
| tanh | 0.2232 (0.0082) | 0.2217 (0.0142) | 0.2396 (0.0068) | 0.2629 (0.0098) |
| relu | 0.2253 (0.0058) | 0.2236 (0.0134) | 0.2436 (0.0104) | 0.2637 (0.0104) |
| $\text{sigmoid}_{mlp}$ | 0.2487 (0.0109) | 0.2681 (0.0140) | 0.2964 (0.0054) | 0.3335 (0.0063) |
| $\text{tanh}_{mlp}$ | 0.2198 (0.0058) | 0.2259 (0.0156) | 0.2358 (0.0024) | 0.2616 (0.0111) |
| $\text{relu}_{mlp}$ | 0.2175 (0.0067) | 0.2263 (0.0144) | 0.2402 (0.0037) | 0.2668 (0.0090) |
| Med2Vec | 0.2162 (0.0091) | **0.2141** (0.0171) | 0.2340 (0.0043) | 0.2631 (0.0106) |
| GRAM | 0.2321 (0.0118) | 0.2291 (0.0154) | 0.2382 (0.0036) | 0.2663 (0.0071) |
| MiME | **0.2128** (0.0075) | 0.2153 (0.0126) | **0.2331** (0.0039) | **0.2559** (0.0096) |
| MiME $_{aux}$ | **0.2111** (0.0089) | **0.2122** (0.0115) | **0.2326** (0.0048) | **0.2557** (0.0095) |

Table 9: Test ROC-AUC of HF prediction for increasing data size. Parentheses denote standard deviations from 5-fold random data splits. Two best values in each column are marked bold.

| | $E_1$ (6299 patients) | $E_2$ (15794 patients) | $E_3$ (21128 patients) | $E_4$ (27428 patients) |
|---|---|---|---|---|
| raw | 0.7585 (0.0202) | 0.8003 (0.0265) | 0.8165 (0.0146) | 0.8330 (0.0111) |
| linear | 0.7411 (0.0252) | 0.7945 (0.0181) | 0.8129 (0.0140) | 0.8377 (0.0119) |
| sigmoid | 0.7236 (0.0286) | 0.7978 (0.0163) | 0.8154 (0.0167) | 0.8343 (0.0121) |
| tanh | 0.7419 (0.0247) | 0.7943 (0.0186) | 0.8121 (0.0146) | 0.8388 (0.0117) |
| relu | 0.7366 (0.0267) | 0.7891 (0.0197) | 0.8105 (0.0210) | 0.8353 (0.0123) |
| $\text{sigmoid}_{mlp}$ | 0.5191 (0.0269) | 0.5356 (0.0365) | 0.6013 (0.0082) | 0.6628 (0.0176) |
| $\text{tanh}_{mlp}$ | 0.7429 (0.0330) | 0.7796 (0.0283) | 0.8172 (0.0084) | 0.8431 (0.0128) |
| $\text{relu}_{mlp}$ | 0.7496 (0.0425) | 0.7837 (0.0217) | 0.8047 (0.0131) | 0.8331 (0.0100) |
| Med2Vec | 0.7633 (0.0151) | **0.8141** (0.0213) | **0.8301** (0.0138) | 0.8445 (0.0115) |
| GRAM | 0.7575 (0.0218) | 0.7828 (0.0228) | 0.8077 (0.0107) | 0.8313 (0.0083) |
| MiME | **0.7676** (0.0292) | 0.8109 (0.0223) | 0.8267 (0.0106) | **0.8471** (0.0100) |
| MiME $_{aux}$ | **0.7824** (0.0213) | **0.8154** (0.0193) | **0.8281** (0.0159) | **0.8478** (0.0108) |

## H  Sequential Disease Prediction

*Sequential disease prediction* In order to test if leveraging EHR's inherent structure is a strategy generalizable beyond heart failure prediction, we test `MiME`'s prediction performance in another context, namely sequential disease prediction. The objective is to predict the diagnosis codes occurring in visit $\mathcal{V}^{(t+1)}$, given all past visits $\mathcal{V}^{(1)}, \mathcal{V}^{(2)}, \ldots, \mathcal{V}^{(t)}$. The input features are diagnosis codes $\mathcal{A}$ and treatment codes $\mathcal{B}$, while the output space only consists of diagnosis codes $\mathcal{A}$. This task is useful for preemptively assessing the patient's potential future risk [10], but is also appropriate for assessing how well a model captures the progression of the patient status over time. We used GRU as the mapping function $h(\cdot)$, and hidden vectors from all timesteps were fed to the softmax function with $|\mathcal{A}|$ output classes to perform sequential prediction.

## I  Experiment Results for Sequential Disease Prediction

Table 10: Prediction performance for sequential disease prediction. Values in the parentheses denote standard deviations from 5-fold random data splits. The best value in each column is marked in bold.

|  | Test loss | Test recall@5 | Test recall@10 | Test recall@20 |
|---|---|---|---|---|
| raw | 7.2121 (0.0319) | 0.5329 (0.0016) | 0.6600 (0.0016) | 0.7749 (0.0019) |
| linear | 7.1474 (0.0321) | 0.5443 (0.0008) | 0.6749 (0.0010) | 0.7876 (0.0009) |
| sigmoid | 7.3494 (0.0438) | 0.5110 (0.0054) | 0.6338 (0.0052) | 0.7529 (0.0029) |
| tanh | 7.1439 (0.0313) | 0.5456 (0.0016) | 0.6755 (0.0012) | 0.7879 (0.0010) |
| relu | 7.1576 (0.0285) | 0.5427 (0.0011) | 0.6716 (0.0016) | 0.7846 (0.0015) |
| sigmoid$_{mlp}$ | 8.7886 (0.0257) | 0.2132 (0.0038) | 0.3466 (0.0031) | 0.5158 (0.0044) |
| tanh$_{mlp}$ | 7.1392 (0.0302) | 0.5470 (0.0010) | 0.6788 (0.0006) | 0.7926 (0.0009) |
| relu$_{mlp}$ | 7.1719 (0.0334) | 0.5433 (0.0010) | 0.6744 (0.0010) | 0.7876 (0.0012) |
| Med2Vec | 7.2429 (0.0283) | 0.5317 (0.0011) | 0.6583 (0.0020) | 0.7752 (0.0016) |
| GRAM | 7.1738 (0.0361) | 0.5390 (0.0016) | 0.6685 (0.0025) | 0.7830 (0.0015) |
| MiME | **7.1224** (0.0326) | **0.5496** (0.0010) | **0.6815** (0.0009) | **0.7945** (0.0014) |

After training all models until convergence, performance was measured by sorting the predicted diagnosis codes for $\mathcal{V}^{(t+1)}$ by their prediction values, and calculating $Recall@k$ using the true diagnosis codes of $\mathcal{V}^{(t+1)}$.

Table 10 shows the performance of all models for sequential disease prediction. `MiME` demonstrated the best performance in all metrics, showing that `MiME` can properly capture the temporal progression of the patient status. It is noteworthy that **linear** displayed very competitive performance compared to the best performing models. This is due to the fact that chronic conditions such as hypertension or diabetes persist over a long period of time, and sequentially predicting them becomes an easy task that does not require an expressive model. This was also reported in [10] where a strategy to choose the most frequent diagnosis code as the prediction showed competitive performance in a similar task.

In order to study whether explicitly incorporating the structure of EHR helps when there are small data volume, we calculated the test performance in terms of $Precision@5$ for predicting each diagnosis (Dx) code of $\mathcal{A}$. In Table 11, we report average $Precision@5$ for four different groups of Dx codes, where the groups were formed by the rarity/frequency of the Dx codes in the training data. For example, the first column represents the Dx codes that appear in the 0.01%-0.05% of the entire visits (433407) in the training data, which are very rare diseases. On the other hand, the Dx codes in the last column appear in maximum 13.39% of the visits, indicating high-prevalence diseases. We selected the best performing activation function **tanh** among the three.

As can be seen from Table 11, except for the rarest Dx codes, `MiME` outperforms all other baseline models, as much as 11.6% relative gain over **tanh**$_{mlp}$. It is notable that **Med2Vec** demonstrated the greatest performance for the rarest Dx code group. However, the benefit of using pre-trained embedding vectors quickly diminishes to the point of degrading the performance when there are at least several hundred training samples.

Overall, `MiME` demonstrated good performance in prediction tasks in diverse settings, and it is notable that they significantly outperformed the baseline models in the more complex task, namely HF

Table 11: Accuracy@5 for predicting diseases grouped by their rarity. The prevalence percentages are calculated by dividing the number of occurrences of each disease by $433407$, the total number of visits in the training data. All values are averaged from 5-fold cross validation.

| Model | 20th-40th percentile (0.01%-0.05% preval) | 40th-60th percentile (0.05%-0.2% preval) | 60th-80th percentile (0.2%-0.8% preval) | 80th-100th percentile (0.8%-13.4% preval) |
|---|---|---|---|---|
| raw | 0.0530 (0.0156) | 0.1907 (0.0128) | 0.2999 (0.0039) | 0.4304 (0.0052) |
| linear | 0.0633 (0.0203) | 0.2162 (0.0163) | 0.3266 (0.0053) | 0.4388 (0.0051) |
| tanh | 0.0674 (0.0182) | 0.2101 (0.0143) | 0.3218 (0.0045) | 0.4379 (0.0033) |
| $tanh_{mlp}$ | 0.0723 (0.0165) | 0.2353 (0.0118) | 0.3388 (0.0044) | 0.4381 (0.0034) |
| Med2Vec | **0.1156** (0.0101) | 0.2240 (0.0155) | 0.3177 (0.0076) | 0.4217 (0.0046) |
| GRAM | 0.0574 (0.0121) | 0.1634 (0.0057) | 0.3053 (0.0089) | 0.4409 (0.0039) |
| MiME | 0.0965 (0.0154) | **0.2625** (0.0209) | **0.3597** (0.0082) | **0.4447** (0.0034) |

prediction, where the relationship between the label and the features (*i.e.* codes) from the data was more than straightforward.