[Reviews · NeurIPS 2018]

Reviewer 1



The authors present a novel methodology designed to take into account some of the structure in the way EHR data is emitted. They observe that a visit consists of a set of diagnosis, which in turn trigger a series of medications and treatments. They leverage this hierarchy to construct multi-level embeddings that allows them to aggregate information across different levels to create embeddings. Overall, I like the idea of using “hidden” structure to improve a model and I like the idea of using the auxiliary loss function. The authors evaluate all methods to predict heart failure under a variety of scenarios. They mention a sequential disease prediction task in the abstract, but I was unable to find an evaluation for this in the main text, though it is mentioned in the supplement. Without commenting on the results for this task in the main the text, I would recommend removing the reference to it. On heart failure, they show that their method improves up Med2Vec and GRAM, two existing methods in this area, especially when data are scarce (Figure 3) and when the visit complexity is high (Table 3). Quality: Over all the paper appears to be sound, though some of the details are missing. I wasn’t clear from the main text or the supplement which version of MiME was the one reported in the main body of the text. Was it the one which included the MLP? If so, as the authors note, this gives MiME and GRAM an advantage of Med2Vec and the fair comparison would add an MLP to Med2Vec. Also, the performance advantage for MiME is less clear when considering the auROC, even though I mostly agree with the authors use of AUC-PR for this purpose. I also had a question with respect to the data source. The authors state: "We conducted all our experiments using EHR data provided by a large healthcare system" More details about the population in this EHR would help the reader assess the generalizability of this data. Is it US? European? What is the socio-economic status of the population? Ethnicity? etc I also know that healthcare data can be hard to come by, but 30,000 patients is not large by any standard. Since most methods performed the same when using all of the data, I am unsure if the results seen by the authors would hold when using an EHR of millions of patients, which is common for most major healthcare centers and insurance claims databases. Clarity: The manuscript is mostly well written, though it is a little hard to follow in spots, mainly in section 2, though I found figure 2 to be very helpful. Small typos: Page 2: In a typical visit,a physician gives a 59 diagnosis to a patient and then order medications or procedures based on the diagnosis. Originality: The method presented here is novel and interesting. Significance: The significance of this work could be improved by using addition datasets (which I know in healthcare is tough) and by considering more conditions. As written, it is of limited significance given the small, elderly population and the single prediction task.

Reviewer 2



The authors present a hierarchical embedding scheme for clinical concept codes with the goal of learning dense numerical representations for forecasting outcomes such as heart failure. The paper is well written and structured clearly. I appreciate the quantitative evaluation campaign (especially using PR over ROC curves). A general comment in this direction is that PR measurements should be treated as point estimates that cannot be simply interpolated (as one would do with ROCs). The introduction hints at the model's potential for application in resource impoverished settings such as detecting rare disorders. While the sub-sampling experiments show promise some robustness to data sparsity, it would be interesting to study the performance on such truly rare outcomes. It would be interesting to additionally conduct a qualitative evaluation, studying not just the extent but also the way in which MIME classification differs from that of state-of-the-art models. A dedicated error analysis would be helpful for understanding any systematic strengths or weaknesses of the respective models. In summary, this is a nice and timely study that presents an effective framework for clinical outcome prediction. Considering the original submission as well as the authors' feedback, I recommend acceptance if space permits.

Reviewer 3



The authors create a multilevel embedding of EHR data (Multilevel Medical Embedding - MiME) that jointly performs prediction tasks where leverage inherent EHR structure should be beneficial - heart failure prediction and sequential disease prediction. The MiME embeddings outperformed the baselines in predicting heart failure Strengths: - The paper is well-written and easy to understand. The authors leverage other work when needed, and emphasize their own contributions. - The experiments are well done - varying data size and visit complexity are good axes to evaluate the embedding on, and MIME is competitive. - MIME has the greatest performance improvement (15% relative gain PR-AUC) on the smallest dataset, which could indicate better out-of-sample performance on unseen hospitals. Weaknesses: - MiME is designed to explicitly capture the relationship between the diagnosis codes and the treatment codes within visits, but this is known to be weak in many cases. E.g., a doctor may give a diagnosis "code" for diabetes, simply so that the patient may be tested for diabetes and have insurance cover the cost. MIME does well here, but I am concerned about the value of this prediction given its artifice. - MIME does best on HF, but not equally well on the other sequential disease prediction task. In the elderly population targeted here, the outcome will obviously be enriched, and doctors will be looking for HF. I am concerned that the performance could be due to learning the overtreatment patterns of healthcare professionals on older patients, especially given that it does so well on the Visit Complexity task, which is actually Dx-treatment interactions.